# The Influence on Post-Activation Potentiation Exerted by Different Degrees of Blood Flow Restriction and Multi-Levels of Activation Intensity

**DOI:** 10.3390/ijerph191710597

**Published:** 2022-08-25

**Authors:** Hang Zheng, Jiajun Liu, Jia Wei, Hui Chen, Shan Tang, Zhexiao Zhou

**Affiliations:** 1Faculty of Sports Science, Ningbo University, Ningbo 315211, China; 2Research Academy of Grand Health, Ningbo University, Ningbo 315211, China; 3Shanghai University of Sport, Shanghai 200438, China; 4School of Strength and Conditioning, Beijing Sport University, Beijing 100084, China; 5School of Sports and Health Management, Chongqing University of Education, Chongqing 400067, China

**Keywords:** post-activation potentiation, recovery time, blood flow restriction, low-intensity, college athletes

## Abstract

(1) Background: To explore the influence on post-activation potentiation (PAP) when combining different degrees of blood flow restriction (BFR) with multi-levels of resistance training intensity of activation. (2) Purpose: To provide competitive athletes with a more efficient and feasible warm-up program. (3) Study Design: The same batch of subjects performed the vertical jump test of the warm-up procedure under different conditions, one traditional and six BFR procedures. (4) Methods: Participants performed seven counter movement jump (CMJ) tests in random order, including 90% one repetition maximum (1RM) without BFR (CON), and three levels of BFR (30%, 50%, 70%) combined with (30% and 50% 1RM) (BFR-30-30, BFR-30-50, BFR-50-30, BFR-50-50, BFR-70-30 and BFR-70-50). Jump height (H), mean power output (P), peak vertical ground reaction force (vGRF), and the mean rate of force development (RFD) were recorded and measured. (5) Results: Significantly increasing results were observed in: jump height: CON (8 min), BFR-30-30 (0, 4 min), BFR-30-50 (4, 8 min), BFR-50-30 (8 min), BFR-50-50 (4, 8 min), BFR-70-30 (8 min), (*p* < 0.05); and power output: CON (8 min), BFR-30-30 (0, 4 min), BFR-30-50 (4 min), BFR-50-30 (8 min), BFR-50-50 (4, 8 min) (*p* < 0.05); vGRF: CON (8 min), BFR-30-30 (0, 4 min), BFR-30-50 (4, 8 min), BFR-50-30 (4 min), BFR-50-50 (4, 8 min) (*p* < 0.05); RFD: CON (8 min), BFR-30-30 (0, 4 min), BFR-30-50 (4 min), BFR-50-30 (4 min), BFR-50-50 (4 min) (*p* < 0.05). (5) Conclusions: low to moderate degrees of BFR procedures produced a similar PAP to traditional activation. Additionally, BFR-30-30, BFR-30-50, and BFR-50-50 were longer at PAP duration in comparison with CON.

## 1. Introduction

Warm-up is not only essential but also necessary to undertake before formal training sessions and races. An effective warm-up can improve athletes’ physical activity and reduce the risk of sports injury. Elite athletes usually use high-intensity warm-ups to enhance their sports performance. Performance is enhanced because of the effect of high-intensity warm-ups on the performance of subsequent explosive activities [1], named muscle post-activation potentiation [2], which is in accordance with the theory of post-activation potentiation (PAP). This activation is usually performed in two ways: static training, which is used for improving explosive power and speed-strength in sports, and resistance and isometric exercises, to make major muscles more pro-active [3]. Two mechanisms have been suggested: (i) the phosphorylation of myosin regulatory light chains and (ii) improved α–motoneuron excitability [4]. Preview studies have proved that 25~100% 1RM resistance exercise will have a positive effect on PAP [5], and the effect of PAP will increase as the activation intensity increases. Previous studies have confirmed the existence of PAP, and it has been observed in many fast movements, including short-distance [6], jumping [3], and throwing [7], and evidence shows it can significantly improve the sports performance of rugby, soccer, weightlifting, sprinting, handball, and swimming athletes [6,7,8,9]. However, in reality, it is difficult to control the application of PAP because of the variety of factors that affect it [10], for instance, training methods, intensity, recovery time, gender, and individual athlete differences. High-intensity activation will also cause severe fatigue, which may lead to sports injuries [1], and the equipment required for high-intensity activation is almost impossible to provide, especially during races and competitions.

Blood flow restriction training (BFRT) has attracted great attention in recent years, because of its efficient characteristics in both strength and aerobic training. Early researchers found similar effects between high-intensity traditional resistance exercise and low-intensity BFRT, with 90% one repetition maximum (1RM) and 20~30% 1RM, respectively [11], and it is currently widely used in many different sports fields [12,13,14,15]. BFRT consists of performing low-intensity resistance training while a relatively light and flexible cuff is placed on the proximal part of an athlete’s lower or upper limb. This provides appropriate superficial pressure for the mechanisms responsible for the positive effects of BFRT [16], including muscle cell swelling, muscle fiber recruitment, systemic hormones (specifically growth hormone, GH), and protein synthesis (MPS) [16,17]. In addition to strength training, benefit was found in BFR aerobic training as well, which is considered to be the concurrent channel of strength and endurance training [18].

In terms of BFRT studies, the main body of research is related to training, with few studies investigating the influence of post-activation during the warm-up. Kim et al. have found the potential of compound muscle action will increase through low-intensity blood flow restriction exercises [17]. Furthermore, a pre-experiment found a significant difference in integral electromyographic (iEMG) between the control leg (without BFR) and the experiment leg (with 50% arterial occlusion pressure (AOP) during the heel raise exercise (experiment leg > control leg; *p* < 0.01) [19]. The study by Wilk et al. [20] shows the PAP effect could occur during the bench press exercise with BFR, and that significant increases in power output and bar velocity during the BP exercise for the BFR condition (70% 1RM with 90% AOP) compared to the control test protocol condition (70% 1RM). These data suggest that resistance exercise with BFR could increase acute physiological responses in muscles, though more substantive research is required to definitively prove the impact of low-intensity blood flow restriction exercises on sports performance. If the combination of BFR with low-intensity blood flow restriction exercises creates the ideal level of PAP, then this will bring great benefits to athletes’ warm-up regimes, as lower-intensity exercises mean a lower risk of injuries, and required team equipment is more convenient to carry and transport.

Considering the selection of appropriate pressure value and resistance load intensity are two important factors of BFRT, which are the key factors leading to the PAP effect. Therefore, this study focuses on blood flow restriction training and aims to investigate the effects on PAP of different degrees of blood flow restriction combined with multi-levels of resistance load intensity, as well as the differences in optimal recovery time of each procedure. It aims to provide a theoretical basis for athletes to easily and efficiently improve speed and explosive performance during warm-up sessions.

## 2. Methods

### 2.1. Participants

Twelve male college athletes volunteered to participate in this investigation (Table 1), from various sports including track and field, basketball, soccer, and badminton. The inclusion criterion was a leg press personal record with a load of at least 200% body mass. They reported resistance training 2 times per week for the past 2 years with experience in the leg press and counter movement jump (CMJ), and without any history of serious lower limbs injuries. All participants were required to refrain from resistance training 72 h prior to each experimental session, and were informed about the benefits and potential risks of the study before providing their written informed consent for participation. The protocol was approved by the ethics committee of the Research Academy of Grand Health, Ningbo University (RAGH2020023302013; January 2020).

### 2.2. General Overview of Experimental Design

The whole experimental design is shown in Figure 1, in order to determine the load intensity and the arterial occlusion pressure value between individuals (BFR groups) in the subsequent experiments, we first measured individual thigh circumference, for the selection of subsequent pressure values [16] and conducted the 1RM leg press test, in accordance with NSCA standards. The experimental part was conducted 7 days after the completion of the 1RM test and 7 rounds of tests were carried out (A: 90% 1RM traditional resistance exercise without BFR, CON; B: 30% AOP with 30% 1RM, BFR-30-30; C: 30% AOP with 50% 1RM, BFR-30-50; D: 50% AOP with 30% 1RM, BFR-50-30; E: 50% AOP with 50% 1RM, BFR-50-50; F: 70% AOP with 30% 1RM, BFR-70-30; G: 70% AOP with 50% 1RM, BFR-70-50). The counter movement jump (CMJ) is a slow stretch shortening cycle (SSC), which is widely used for research purposes in the applied setting, therefore, the CMJ has been employed for this study to evaluate the performance. The order of the rounds was randomized, and the recovery period between the 7 rounds was at least 48 h.

### 2.3. Experimental Sessions

For the formal experiment, all rounds began with a 10-min standardized warm-up, in accordance with NASA standards [21], which comprised 5 min of jogging, and then a set of dynamic lower body activation exercises. Next, subjects completed a baseline CMJ, then proceeded onto an activation exercise on the leg press machine (all rounds were performed with a set of 5 leg press repetitions, with a 60-s interval between the repetitions). Immediately after the activation (within 15 s) and every 4 min after the activation for up to 12 min, the subjects repeated the CMJ 3 times consecutively. The 4-min interval was to avoid warm-up or fatigue effects during the main experiment, and verbal encouragement was given to ensure the subjects’ best performance.

To obtain the kinematics and dynamics data of each jump, the CMJ was completed on a portable force platform (Kistler Quattro Jump). Every jump was performed under the guidance of the commanding staff member and the platform reset every time a subject finished, before then importing the next subject’s information (body mass, height, etc.). The subject’s body mass was first measured for the calculation of the platform formula. The subject then got ready with arms akimbo on the platform, and completed the CMJ after the staff member said the codeword “jump”. It took approximately 3 s for the platform to record and save the jump data and the bout was completed after three jumps.

The changes in jump height (cm), average power output (Watt), vertical ground reaction force (N), and center of gravity (N/s^−1^) were obtained directly from the plate system, whereas the initial data of the fore during the performance of the CMJ were given by the multiple of body mass. The formula of peak vertical ground reaction force (vGRF) is defined as: multiple of body mass (MBW) × body mass (kg) × gravity (*g* = 9.8 m/s^−2^). The average rate of force development is equal to the peak vertical ground reaction force divided by the time to reach maximum force (T) and represents the average rate of force development during take-off. The specific formula of the average rate of force development (RFD) is based on vGRF/T [22].

### 2.4. Statistical Analysis

After testing the data for normal distribution, data were expressed at the Mean ± SD. A 7 × 5 (treatment × time) and repeated measures analysis of variance were performed to estimate the significant differences between groups, as well as the indicators at the five-time points of each protocol. When significant *F*-values were observed (*p* ≤ 0.05), paired comparisons were used to determine significant differences, in conjunction with Holm’s Bonferroni method for control of type 1 error. The level of significance was set at *p* ≤ 0.05 in this study, and all statistics were performed using SPSS 24.0 (SPSS Inc., Chicago, IL, USA).

## 3. Results

The repeated measures ANOVA revealed a significant time effect on jump height (*F* = 17.316, *p* = 0.000, $\eta_{p}^{2}$ = 0.184), while no significant effect was observed in the different groups (*p* > 0.05), and no significant interaction was found (*p* > 0.05). With specific time points in the experimental groups, a significant difference was observed (*p* = 0.041) when compared with the 8 min of CON (52.03 ± 4.06 cm) to the baseline (50.46 ± 4.12 cm). For the BFR groups, the performance showed significant higher jump height when compared with baseline: BFR-30-30 (baseline: 50.18 ± 4.6 cm, 0 min: 52.12 ± 4.22 cm, *p =* 0.048; 4 min: 52.48 ± 3.81 cm, *p* = 0.043), BFR-30-50 (baseline: 50.58 ± 5.43 cm; 4 min: 52.98 ± 4.61 cm, *p* = 0.036; 8 min: 52.54 ± 4.82cm, *p* = 0.049), BFR-50-30 (baseline: 50.06 ± 4.15 cm; 8 min: 52.21 ± 4.46 cm, *p =* 0.044), BFR-50-50 (baseline: 50.19 ± 3.92 cm; 4 min: 52.31 ± 4.52 cm, *p* = 0.041; 8 min: 51.83 ± 4 cm, *p* = 0.049) and BFR-70-30 (baseline: 50.23 ± 5.38 cm; 8 min: 53.1 ± 3.54 cm, *p* = 0.048) (Figure 2).

In terms of the average power output, a repeated measures ANOVA revealed a significant time effect over the duration of this study (*F* = 4.946, *p* = 0.009, $\eta_{p}^{2}$ = 0.31), for the specific time points, significant changings were observed in five of the seven groups: CON (baseline: 2056.4 ± 383.5 W; 8 min: 2210.59 ± 350.87 W, *p* = 0.015), BFR-30-30 (baseline: 2062.8 ± 422.66 W; 0 min: 2162.55 ± 417.07 W, *p* = 0.026; 4 min: 2170.5 ± 384.98 W), BFR-30-50 (baseline: 2052.42 ± 316.94 W; 4 min: 2180.4 ± 327.43 W, *p* = 0.042), BFR-50-30 (baseline: 2007.81 ± 340.54 W; 8 min: 2126.37 ± 325.04 W, *p* = 0.034), BFR-50-50 (baseline: 2065.07 ± 368.59 W; 4 min: 2153.64 ± 391.51 W, *p* = 0.042; 8 min: 2112.71 ± 409.84 W, *p* = 0.017), while no difference was observed in the 70% AOP groups (Figure 3).

vGRF showed similar results to the Power output, a significant time effect was observed through the analysis (*F* = 8.707, *p* = 0.001, $\eta_{p}^{2}$ = 0.442). For the CON, the vGRF at 8 min (1824.22 ± 304.37 N) is significant higher than that at the baseline (1699.08 ± 290.15 N, *p* = 0.043), the vGRF at 0 and 4 min of BFR-30-30 was significant higher than the baseline (baseline: 1658.36 ± 336.75 N; 0 min: 1837.12 ± 291.68 N, *p* = 0.027; 4 min: 1819.91 ± 260.8 N, *p* = 0.044), BFR-30-50 (baseline: 1610.6 ± 262.84 N; 4 min: 1836.71 ± 316.66 N, *p* = 0.026; 8 min: 1842.7 ± 295.81 N, *p* = 0.012), BFR-50-30 (baseline: 1667.21 ± 284.24 N; 4 min: 1850.74 ± 331.02 N, *p* = 0.019), BFR-50-50 (baseline: 1658.5 ± 335.01 N; 4 min: 1852.56 ± 332.62 N, *p* = 0.046; 8 min: 1817.75 ± 309.34 N, *p* = 0.039), and no significant difference was observed in the 70% AOP groups as well (Figure 4).

In terms of RFD, different groups showed main effect (*F* = 2.747, *p =* 0.019, $\eta_{p}^{2}$ = 0.2), and different time points displayed main effects (*F* = 8.331, *p =* 0.000, $\eta_{p}^{2}$ = 0.43), and significant interaction (*F* = 2.973, *p =* 0.000, $\eta_{p}^{2}$ = 0.213). With following further analysis of the specific groups, the RFD at 8 min of CON was significant higher than baseline (baseline: 6003.38 ± 1175.22 N/S; 8 min: 6769.68 ± 1302.86 N/S, *p =* 0.01), BFR-30-30 (baseline: 5901.8 ± 1592.35 N/S; 0 min: 6894.78 ± 1592.35 N/S, *p =* 0.005; 4 min: 6942.57 ± 1217.82 N/S, *p =* 0.002), BFR-30-50 (baseline: 5593.05.8 ± 1075.81 N/S; 4 min: 6604.58 ± 1035.02 N/S, *p =* 0.033), BFR-50-30 (baseline: 5914.43 ± 1486.22 N/S; 4 min: 6666.42 ± 1402.14 N/S, *p =* 0.029), BFR-50-50 (baseline: 5716 ± 1455.61 N/S; 4 min: 6445.62 ± 1234.67 N/S, *p =* 0.034), and no difference was found in the 70% AOP groups. Between the seven groups of each time point, the RFD at 0 min of BFR-30-30 (6894.78 ± 1354.29 N/S) displayed significant higher RFD than that at 0 min of CON (6002.97 ± 956.98N/S, *p =* 0.035), BFR-50-30 (5912.28 ± 1197.17 N/S, *p =* 0.035), BFR-50-50 (5986.42 ± 1484.06 N/S, *p =* 0.019), BFR-70-30 (5599.68 ± 1194.09 N/S, *p =* 0.002) and BFR-70-50 (5474.28 ± 1029.81N/S, *p =* 0.002). At the time point of 4 min, BFR-30-30 (6942.57 ± 1217.82 N/S) displayed significant higher RFD than BFR-70-30 (5768.24 ± 1294.05 N/S, *p =* 0.026) and BFR-70-50 (5611.15 ± 1178.31 N/S, *p =* 0.003), BFR-50-30 (6666.42 ± 1402.14 N/S) was significant higher than BFR-70-30 (*p =* 0.013) and BFR-70-50 (*p =* 0.003), lastly, BFR-50-50 (6445.62 ± 1234.67N/S) was significant higher than BFR-70-30 (*p =* 0.041) and BFR-70-50 (*p =* 0.018) (Figure 5).

## 4. Discussion

The results demonstrate that a low to moderate degree of blood flow restriction combined with low to moderate intensity resistance exercises as activation will produce a similar PAP to traditional high-intensity activation during warm-up. Furthermore, the results indicated that, when compared with traditional high-intensity exercises, specific BFR activation will produce a longer duration of PAP.

The primary aim of this study was to investigate the effect of using BFRT as an activation exercise, and the difference in PAP in the following recovery periods between the seven experimental groups. In the present study, a significant improvement in CMJ performance was observed in CON, BFR-30-30, BFR-30-50, BFR-50-30, BFR-50-50, and BFR-70-30 through the jump height. For the traditional group, previous studies have confirmed the existence of PAP after traditional high-intensity activation [5,7]. To explain it from the perspective of mediated physiological mechanism, the phosphorylation of myosin regulatory light chains (RLC) was responsible [23], and relevant evidence showed that the phosphorylation of RLC played an important role in potentiation, which is affected by the released Ca^+^ molecules from the sarcoplasmic reticulum during muscular contraction [23], and the subsequent contractions will be potentiated through this altering. In addition, previous findings have shown that low-intensity BFRT can also yield a significant increase in Ca^+^ [24]. In this study, there was no significant difference between each group’s best performance at any following time point after the activation, which indicates that low-intensity BFRT activation is also theoretically able to produce PAP. Therefore, though further investigation is needed, this study speculates it to be one of the mediated mechanisms in the potentiation of BFR groups.

To estimate explosive performance during jump “take off”, power output, vGRF, and RFD were selected and recorded during the experiment. On the issue of improving athletes’ power output, previous studies have pointed out that 40–70% 1RM upper limb [25] and 60% 1RM lower limb [8] resistance exercises are more reliable, whereas, after further research, it is now generally believed that ≥80% 1RM resistance exercises are more efficient and can produce the best power output during subsequent performance [9]. Another interpretation for PAP is that a maximal or a near-maximal heavy resistance intensity will increase motor unit activation [23]. As the present results showed, significant changes in Power, vGRF and RFD were observed in CON, verifying the views of previous researchers. While changes in these indexes were also found in the majority of BFR groups, four of the six BFR groups had significant changes in vGRF and RFD, and BFR-30-30 reached the highest RFD out of the seven groups, with similar results appearing for vGRF. In contrast, there was no significant difference in vGRF and RFD between the following recovery times after activation in the 70% AOP groups. RFD is a key index in estimating the fiber power output during exercise. Relative research has reported that RFD will increase as the neural drive enhances, especially during plyometrics [26]. Kilduff reported that CMJ height, power, and RFD were significantly higher at 8 min in rugby athletes after traditional high-intensity activation [3], and previous evidence has reported that MVC stimulation has a significant effect on H-wave changes in human muscle fibers, with the author explaining that it is probably due to changes in the recruitment ability of higher-order motor units [27].

Before training sessions or races, it is vital to carry out a warm-up. Warm-ups reduce the risk of sports injuries and activate both the higher-order motor units in muscle fibers and the nervous systems to improve the athletes’ performance in subsequent activities. In earlier studies of BFR, evidence reported that the high-order motor unit is not only related to the force and speed of muscle contraction but is also related to oxygen concentration during exercise [28]. In addition to restricted blood flow, there was also a significant reduction in tissue oxygen saturation during BFRT [29], which is correlated to the high-order motor units. To verify this finding, relative studies were collected and analyzed in this study. The findings showed a greater iEMG in the BFRT group when compared with the traditional training group (*p <* 0.05) [17], which supported the theory of muscle fiber recruitment during BFR groups. Moreover, the changes in muscle fiber pennation angle may be another cause of PAP during the activation exercises [30]. These changes are thought to affect the force transmission from tendons and bones, which means smaller pennation angles will have a mechanical advantage in force transmission to the tendon [31]. Mahlfed reported that isometric exercises are effective in reducing fiber pennation angle, and are thought to be responsible for producing PAP [30]. In terms of BFRT, evidence also showed that when compared with the high-intensity traditional resistance training, low-intensity BFRT will have a similar effect on the pennation angle during exercise [32]. According to biomechanics, a higher transmission efficiency during human physical activity is produced with a smaller pennation, therefore, this study believes this could be one of the causes of PAP in BFR groups.

Acute increases in performance can be substantially affected by the balance between PAP mechanisms and fatigue [33,34]. In general, the higher the intensity, the longer the rest interval the athlete needs to dissipate fatigue [35]. A meta-analysis indicated larger effect sizes of PAP occurring for athletes compared with nonathletes [36]. Athletes may have augmented responses to PAP protocols due to training adaptations, such as increased fatigue resistance. The factor of recovery time had a significant influence on the subsequent performance of the majority groups in this study, as it is shown in Figure 2. With regards to the question of choosing the optimal recovery time after activation, previous studies suggested a range from 4 to 12 min [3,5,9], and most of the studies confirmed 4 min [9] and 8 min [3,6] to be the optimal recovery time after high-intensity PAP activation. In our study, we observed a decrease in jump height, power, vGRF, and RFD when the CMJ was performed immediately after the activation in CON, which is in agreement with the previous literature [3]. Unlike CON, an earlier appearance of PAP was observed after activation in three BFR groups (BFR-30-30: immediately after the activation, BFR-30-50 and BFR-50-50: 4 min after the activation), this suggests that specific BFRT activation may induce less fatigue, which can produce PAP effects through a shorter recovery time. In terms of BFR strength training, studies have typically investigated the effects of fatigue during exercise, and the evidence has shown that continuous BFR causes higher physiological and metabolic stress [32], as well as fatigue [37]. In comparison with interval BFR, there is a greater impact in continuous BFR during exercise, which is applied by the present study. As other evidence has reported that BFRT fatigue only lasts a short time [38], therefore, we safely came to the hypothesis that specific BFR groups in the present study exert a minimal influence on fatigue and that the effects of potentiation played a more important role during the whole process and ultimately improved performance.

The main findings of this study are that the combination of 30% or 50% AOP with 30% or 50% 1RM seems to be a more efficient way of producing PAP. As we can see from Figure 2, a longer duration of PAP was observed in several BFR groups when compared with the short-lived PAP in CON, the 4-min continuous PAP in the three BFR groups indicated that the improvement of athletes’ command ability was more stable, while PAP appeared later in CON required a longer waiting time. The “traditional potentiation-fatigue model” of previous studies and the “hypothetical potentiation-fatigue Model” of this study are shown in Figure 6 below. It is known that races do not usually begin as soon as the athletes finish the warm-up, there is normally a couple of minutes interval between the end of the warm-up and the start of the race. The larger PAP window makes it easier for athletes to perform at their best, because in track and field events, especially in the 100 m, a difference of 0.01 s can decide the winner of a race. Furthermore, in the high jump competition, to reach a certain height, athletes may need to take continuous challenges in a short time due to the restriction of the rules. In addition, previous studies have shown that with different forms of activation, rowers perform better in 3 min of all-out rowing [39]. Combined with the results of this experiment, we suggest that, theoretically, a larger window may benefit athletes in short to medium-distance endurance events (time < 4 min).

Referring to the extent of blood flow restriction, we observed that only low to moderate BFR combined with low to moderate intensity significantly improved PAP, as evidenced by the improvement of jump performance in BFR-30-30, BFR-30-50, BFR-50-30, and BFR-50-50. This result indicates excessive blood flow restriction has a negative influence on muscle fiber activation and may cause a reduction in explosive performance. In strength training, as reported by previous studies, it is not appropriate to use a high degree of arterial occlusion pressure, otherwise, there will be no more further improvement in muscle activation, strength, mass [41], and even negative responses when AOP ≥ 70% [32]. Corresponding conclusions are also drawn in the PAP activation of this study. For the 70% AOP groups, only jump height increased significantly at 8 min of BFR-70-30, this means a high degree of AOP during BFR activation has little effect on PAP. Previous studies suggest that a moderate degree of AOP is better than a high degree for both strength [41] and aerobic exercise [29]. Loenneke investigated the influence of different degrees and intensities of AOP on the activation of muscle fiber, and the results showed that muscle activation increases from 40~50% AOP, but does not increase further with a higher AOP (50~60% AOP), thus showing that high degree of AOP does not greatly affect the torque [41]. This study’s findings support the literature as optimal AOP was found to exist in a range from 30~50%, and displayed a better outcome in PAP when compared with a more than 50% AOP activation.

In reality, there is a high risk of injuries during high-intensity activation exercises, and equipment is much more difficult to carry and transport than low-intensity BFR procedure equipment. What is of greater concern here, however, it is necessary to take the various factors such as age, training history, strength level of the subjects and individual differences, as well as environmental conditions such as the location, weather, and laboratory environment, into account, as they are important to the outcome of PAP. Therefore, further research into the theory of low-intensity BFRT activation for warm-up is needed.

## 5. Limitations

Two limitations of this study must be acknowledged. The first limitation is that the number of subjects is slightly insufficient, so the results may be biased by the fact that we only have a small sample size of males practicing different sports, thus, the practical implication section should be read with caution. Another limitation of this study is that we did not monitor changes in cardiovascular parameters (e.g., heart rate, blood pressure) during the different levels of BFR. Some studies [42,43] have shown that heart rate and blood pressure in BFRT are higher when compared to the control condition, therefore, we hope further studies can focus on these variables.

## 6. Conclusions

The present study demonstrates that the specific activation of BFRT procedures has a significant effect on PAP when applied in the warm-up section. Moreover, PAP appears earlier and lasts longer than that in traditional high-intensity procedures, this means that the specific BFRT activation can not only increase PAP but also avoid the negative effects of high-intensity or high-load activation, which indicated that athletes in need can perform 5 min BFR-30-30, BFR-30-50, and BFR-50-50 warm-up 5 to 10 min right before the training or races for activation, their choices depend on the demands they required.

## 7. Practical Implications 

(1)The 30–50% BFR combine with 30–50% 1RM resistance exercise can replace 90% 1RM traditional exercise as a means of activation during warm up section.(2)Athletes in need can perform BFR-30-30 before the training or race is about to start, or finish BFR-30-50, BFR-50-50 4 min before for activation, their choice should according to the time interval between the end of warm-up and the start of the race.(3)The longer PAP duration after the specific BFRT activation procedure is more reliable for athletes to maintain high level sports performance.(4)Competitive teams should encourage BFRT for activation to reduce the transportation burden on the team during the competition.

## Figures and Tables

**Figure 1 ijerph-19-10597-f001:**
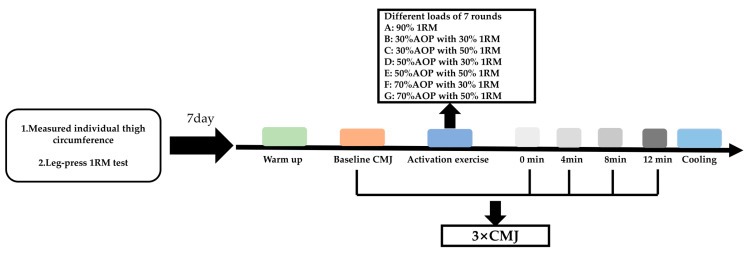
The experimental design.

**Figure 2 ijerph-19-10597-f002:**
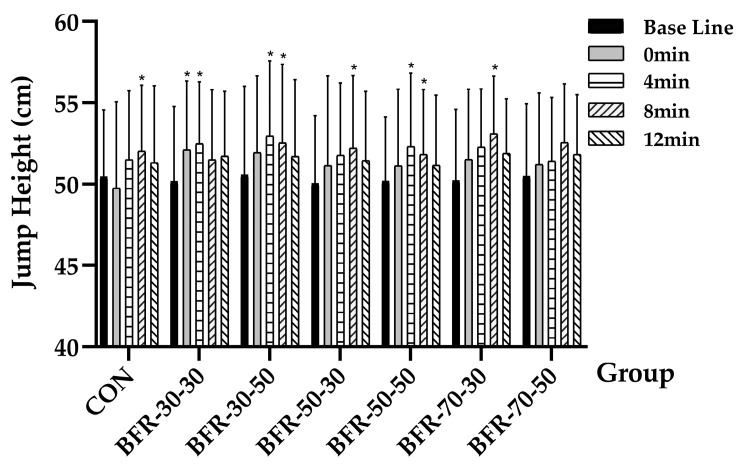
Jump height during countermovement jumps before and after the activation. “*” indicates a significant increase compared with baseline (*p* < 0.05).

**Figure 3 ijerph-19-10597-f003:**
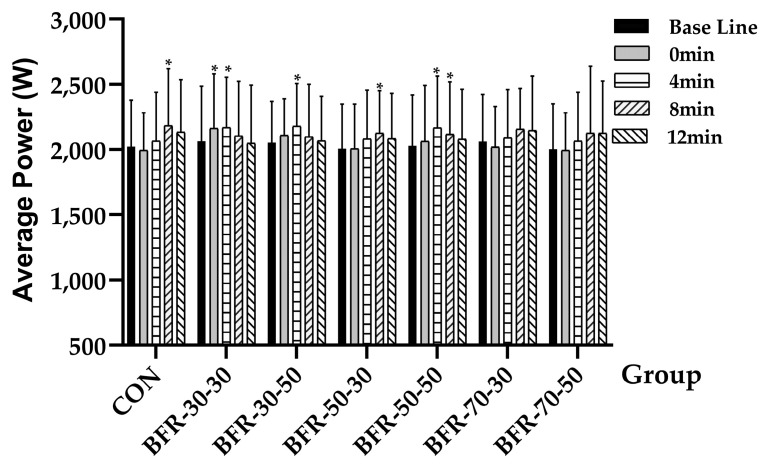
Average power output during countermovement jumps before and after the activation. “*” indicates a significant increase compared with baseline (*p* < 0.05).

**Figure 4 ijerph-19-10597-f004:**
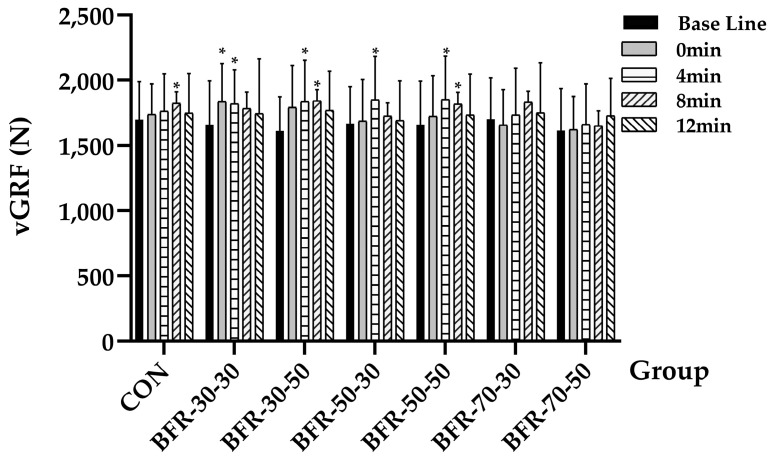
Peak vertical ground reaction force during countermovement jumps before and after the activation. “*” indicates a significant increase compared with baseline (*p* < 0.05).

**Figure 5 ijerph-19-10597-f005:**
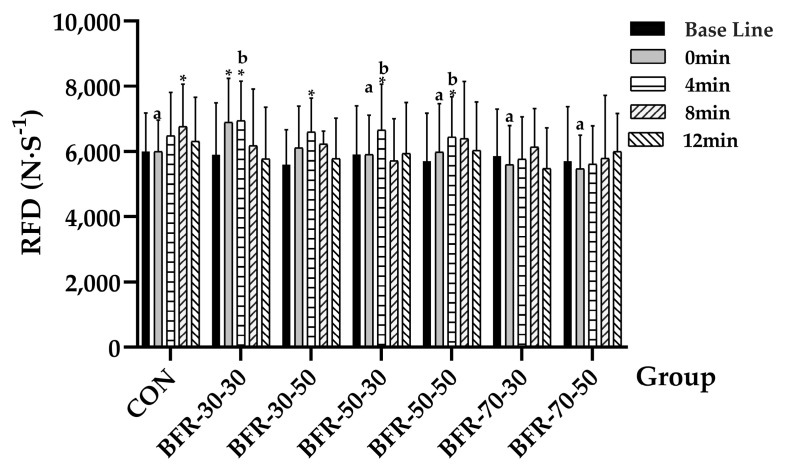
Average RFD during countermovement jumps before and after the activation. “*” indicates a significant increase compared with baseline (*p* < 0.05). “a” means significantly lower than the value at 0 min of BFR-30-30, and “b” means significantly higher than the value at 4 min of BFR-70-30 and BFR-70-50 (*p* < 0.05).

**Figure 6 ijerph-19-10597-f006:**
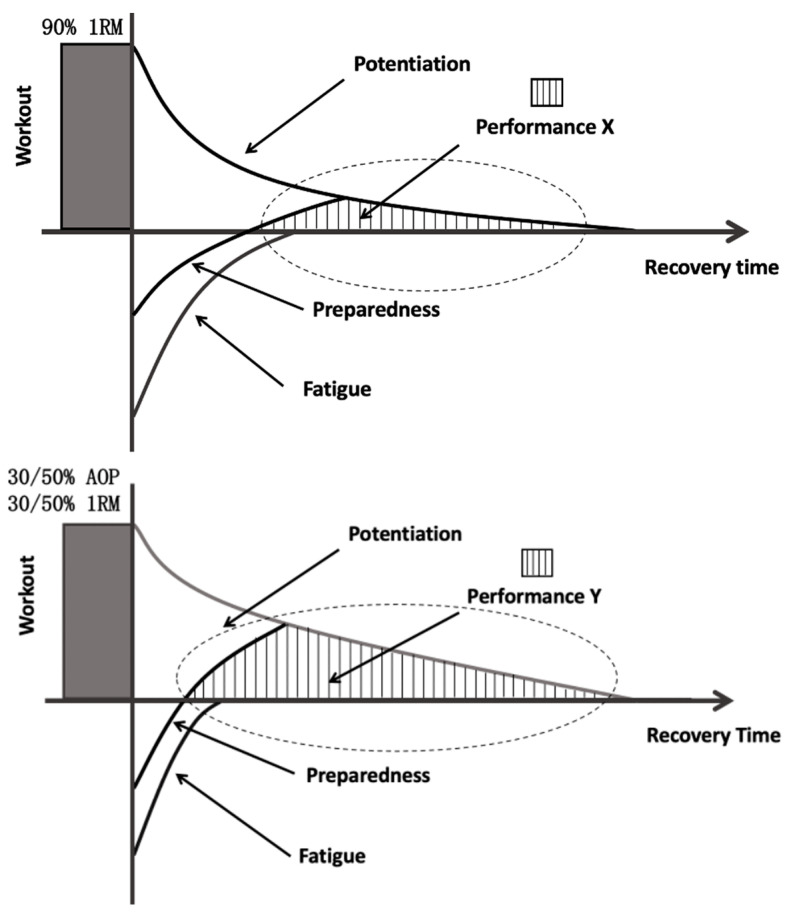
Traditional Potentiation-Fatigue model [40] and BFR Potentiation-Fatigue Model (Speculations from this study, BFR means 30% or 50% BFR combine with 30% or 50% 1RM during activation). The figure adapted from Owen, W. Post-Activation Potentiation. Science for Sport, 2016. Available online: https://www.scienceforsport.com/post-activation-potentiation/ (accessed on 21 July 2022).

**Table 1 ijerph-19-10597-t001:** Physical characteristics of subjects at baseline, *n* = 12.

Variables	Means ± SD
Age (y)	22.7 ± 2
Height (m)	1.76 ± 3.82
Mass (kg)	66.7 ± 8.34
BMI (kg/cm^−2^)	22.35 ± 4.03

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
