# Peer review of "The Influence on Post-Activation Potentiation Exerted by Different Degrees of Blood Flow Restriction and Multi-Levels of Activation Intensity"

_ijerph, 2022, doi:10.3390/ijerph191710597_

Round 1
Reviewer 1 Report
The manuscript is original, has relevant theoretical background and it is well written. The findings are interesting in terms of applicability in the sports field. There are some few reflections below that should be considered.
1) There are several acronyms that were not described in the abstract and throughout the text, which makes reading a little difficult.
2) The description of the experimental protocol is a bit confusing and difficult to understand. They involve different protocols and different time windows between the different experiments (i.e., 1RM, 7 rounds of tests, CMJ assessment, leg press exercise…). I suggest the authors include a Figure with the timeline and description of the experiments in order to make it easier to understand the order, procedures, intervals, evaluations, etc.
3) Although this is not the objective of the study, it would be interesting, for the authors to include data related to the cardiovascular parameters (e.g., Heart Rate, Blood Pressure) monitored during the different levels of BFR, to provide support regarding the safety of the protocols, if the authors have monitored these variables.
4) The authors should better explain and provide a theoretical basis for the choice of the experimental protocol, that is, what is the rationale why the authors chose this wide range of intensities of 1RM and vascular occlusion, as well as their combinations?
5) I suggest the authors discuss a little more in depth about the effects of PAP duration after the specific BFR activation during warm-up on sports performance in different sports modalities.
Author Response
Dear reviewer,
First of all, on behalf of all authors, I would like to many thanks to the expert who put forward valuable advice! According to the modification opinions of the experts, after careful discussion and understanding again, the revision will be made one by one, and the modification description is reported as follows:
Point 1: There are several acronyms that were not described in the abstract and throughout the text, which makes reading a little difficult.
Response 1: we checked the manuscript to make sure the special noun had been described before it was used as an acronym.
Point 2: The description of the experimental protocol is a bit confusing and difficult to understand. They involve different protocols and different time windows between the different experiments (i.e., 1RM, 7 rounds of tests, CMJ assessment, leg press exercise…). I suggest the authors include a Figure with the timeline and description of the experiments in order to make it easier to understand the order, procedures, intervals, evaluations, etc.
Response 2: As you suggested, we have added a figure of the experimental design in the manuscript to help readers understand the order, procedures, intervals, evaluations, etc.
Point 3: Although this is not the objective of the study, it would be interesting, for the authors to include data related to the cardiovascular parameters (e.g., Heart Rate, Blood Pressure) monitored during the different levels of BFR, to provide support regarding the safety of the protocols if the authors have monitored these variables.
Response 3: After reading your suggestion, we also realize that this may be an additional influencing factor. Unfortunately, we did not collect cardiovascular parameters in our study. We put this in the limitations section and hope that other auth will take note.
Point 4: The authors should better explain and provide a theoretical basis for the choice of the experimental protocol, that is, what is the rationale why the authors chose this wide range of intensities of 1RM and vascular occlusion, as well as their combinations?
Response 4: In the PAP study, we found that the weight of resistance exercise load was an important influencing factor, while in the BFRT study, the pressure value of pressure was an important influencing factor. Therefore, we need to consider both factors in this study.
Point 5: I suggest the authors discuss a little more in depth about the effects of PAP duration after the specific BFR activation during warm-up on sports performance in different sports modalities.
Response 5: We have reworked the discussion section with appropriate revisions. You can review the specific changes in the manuscript.
Reviewer 2 Report
The authors made an interesting research to check the effects of different BFRT in athletes. However, there are some indications that may be take into account before publishing this study.
Please check that the journal allows abbreviations without explaining them first at the abstract.
The introduction is well organized and allows the reader to understand the context of the research. I would reccomend the authors to check that all abbreviations are well indicated. Although the authors indicate that the application of blood flow restriction training has increased in the last years, the references are not really updated. Thus, I think introduction could be enhanced by more recent literature and randomized controlled trials applying BFRT.
In the methods, why only 12 participants? I think whis may be a bias on your research. Moreover, the participants are from different specialties as mentioned in your manuscript, which may also induce a bias on the results of your study. You do not mention any inclusion or exclusion criteria, nor how they were recruited.
*Line 100, "NASA standards" Do this need a reference?
*Line 102; what is CMJ? this is not explained before in the text.
Results are crearly described, although the figures are a bit confusing. Could you use different patterns on the groups?
In the discussion, you start with the physiological mechanism of the PAP, although you have not mention this physiological mechanism on the introduction. I would at least make a brief comment about this on the introduction, so the reader can link this information to the information previously included on the manuscript.
*Line 215, you say "recent studies"... but this is related to a reference dated on 1995. I don't think this is recent at all to make this statement.
From line 263 and so on, your discussion seems part of the results. Please try not to talk about your results that much and try con discuss them with previous literature or justify them.
I would reccommend to add a limitations section including the fact that your results may be biased by the fact that you have a small sample saze of males practising different sports. Thus, the practical implication section should be read with caution.
Reviewer 3 Report
This article examines the enhancement of muscle post-activation potentiation performance in athletes with varying degrees of blood flow restriction combined with multi-level resistance training. This study has important implications for developing a more efficient and feasible warm-up program for athletes. But there are still several questions that need to be answered or revised before acceptance.
1. Line 69 The definition of AOP is suggested to explain the meaning in the introduction, which is convenient for more readers to understand.
2. According to Figure 2, the 0 min and 4 min of BFR-30-30 have significant changings, which do not match the description in line 156.
3. The 8 min of BFR-50-50 has no significant changings in figure 2, which does not match the description in line 159.
Author Response
Dear reviewer,
First of all, on behalf of all authors, I would like to many thanks to the expert who put forward valuable advice! According to the modification opinions of the experts, after careful discussion and understanding again, the revision will be made one by one, and the modification description is reported as follows:
Point 1: Line 69 The definition of AOP is suggested to explain the meaning in the introduction, which is convenient for more readers to understand.
Response 1: AOP is arterial occlusion pressure, we checked the manuscript to make sure the special noun had been described before it was used as an acronym.
Point 2: According to Figure 2, the 0 min and 4 min of BFR-30-30 have significant changings, which do not match the description in line 156.
Response 2: We checked the figure again, the 0 min and 4 min of BFR-30-30 have significant changes, then we have corrected them in the manuscript.
Point 3: The 8 min of BFR-50-50 has no significant changings in figure 2, which does not match the description in line 159.
Response 3: We checked the data again, the 8 min of BFR-50-50 has significant changings, then we have corrected the figure.
Round 2
Reviewer 2 Report
Authors have done all changes reccomended by reviewers. In my opinion, this paper is ready to be published